# An Intermodulation Radar for Non-Linear Target and Transceiver Detection

**DOI:** 10.3390/s24051433

**Published:** 2024-02-23

**Authors:** Stefano Pisa, Alessandro Trifiletti, Pasquale Tommasino, Pietro Monsurrò, Piero Tognolatti, Giorgio Leuzzi, Alessandro Di Carlofelice, Emidio Di Giampaolo

**Affiliations:** 1Department of Information Engineering, Electronics and Telecommunication, Sapienza University of Rome, 00184 Rome, Italy; alessandro.trifiletti@uniroma1.it (A.T.); pasquale.tommasino@uniroma1.it (P.T.); pietro.monsurro@uniroma1.it (P.M.); 2Dipartimento di Ingegneria Industriale e dell’Informazione e di Economia, Università degli Studi dell’Aquila, 67100 L’Aquila, Italy; piero.tognolatti@univaq.it (P.T.); giorgio.leuzzi@univaq.it (G.L.); alessandro.dicarlofelice@univaq.it (A.D.C.); emidio.digiampaolo@univaq.it (E.D.G.)

**Keywords:** non-linear target, intermodulation radar, transceiver, electronic devices

## Abstract

The design and the characterization of a non-linear target to test an intermodulation radar was performed using the AWR design environment Version 22 by Cadence software. Two experimental setups for intermodulation measurements were realized in order to characterize connectorized or antenna-equipped devices. Both setups were modeled using the VSS software available inside AWR Version 22. The comparison between measurements and simulations on the designed target showed a very good agreement. Intermodulation measurements were performed on connectorized devices present inside electronic systems and on various transceiver available on the market. This experimental study evidenced that the non-linearities of devices such as amplifiers and mixers are visible at their access ports even when the device is switched off. Moreover, this study highlights the ability of an intermodulation radar to remotely detect the presence of a particular transceiver, even when the latter is switched off, thanks to the specific frequency response of its intermodulation products.

## 1. Introduction

Intermodulation distortion (IMD) is generated in a non-linear device when two tones (f1 and f2) close in frequency are injected into it [1]. Intermodulation can be theoretically predicted using a Volterra series of the device characteristic, or more approximately by a Taylor series [1]. The spectrum at the output of a non-linear device not only consists of the original signals, but also contains other frequency tones, among which the third-order intermodulation (IM3) products (2f1–f2 and 2f2–f1) can cause interference as they are close to the original signals. Moreover, until saturation, the output power of the IM3 increases by 3 dB for each dB of input variation, and therefore much more rapidly than the power of the injected tones which grows by 1dB/dB [1]. A peculiar characteristic of the IM3 products is their asymmetry as the two generated tones have different amplitudes. In [2,3] it was demonstrated that this phenomenon is due to the termination impedances at the difference frequency (f2–f1). The generation of intermodulation is an adverse phenomenon in electronic devices and systems as it determines a broadening of the spectrum of a modulated signal (spectral regrowth), which generates interferences in adjacent channels. However, this effect can be used constructively in radars to detect non-linear targets. In particular, an intermodulation radar utilizes its transmitting antenna to radiate two close-in-frequency tones towards the target. If the target is equipped with an antenna, and contains non-linear elements, intermodulation tones are produced. Of these, those of the third order are close in frequency to the fundamental ones and therefore are re-radiated by the target antenna. The radar-receiving antenna captures these signals and sends them towards the receiving section where they are measured. In laboratory realizations, the receiving section can be simply implemented using a spectrum analyzer [4]. This basic structure can be optimized by inserting amplifiers, to increase the output power, and circulators and diplexers, to minimize the spurious intermodulation produced by the elements of the system and by the receiver [5].

Intermodulation radars have been used for the remote sensing of physiological activities [5,6,7,8]. In [6], the intermodulation response of a Schottky diode connected to an antenna placed on a phantom allowed the remote study of the movement of the phantom even in the presence of strong environmental clutter in a condition where a conventional radar was unable to detect movement. With an intermodulation radar operating around the frequency of 5.8 GHz and using a non-linear target consisting of an array of 36 patch antennas, loaded with Schottky diodes, placed on the chest of a subject in correspondence with the heart, it was possible to record the cardiac activity of the subject isolating it from the more intense breathing signal [5]. Furthermore, using as the target a dipole antenna loaded with a Schottky diode placed on a subject’s wrist at the radial artery, it was possible to monitor the subject’s heart rate [7]. With an intermodulation radar it is also possible to evaluate the distance of a target. In [7], an experimental setup was developed in which the first tone f1 is kept constant and the second tone is switched between two frequencies, f2L and f2H (frequency-shift-keying—FSK method). In this way the target distance was evaluated by measuring the phase difference of the intermodulation tones. Similar results were also obtained by frequency modulating (FMCW) the second tone [8].

Another application of intermodulation radars is the detection of commercial RF electronic devices. In particular, these radars can be used to detect hostile electronic devices that can be remotely activated by an operator via radio frequency transmitters [9]. Additionally, they can help law enforcement officers to locate devices whose emissions exceed those permitted by law, and can allow security personnel to detect unauthorized radio electronics in restricted areas [10]. Finally, they can be used by first responders to locate personal electronics during emergencies such as immediately after an avalanche or earthquake [11,12]. In [9,13,14], by using an intermodulation radar operating between 300 MHz and 2500 MHz, the authors were able to detect the presence of several electronic devices, highlighting the operating band of the various targets. In [15], it was evidenced that devices with the same design and layout present different intermodulation responses. Therefore, the intermodulation produced by a device can also be used to distinguish an original device from a counterfeit one. A two-tone test was used to study the susceptibility to third-order intermodulation of radiofrequency front-ends [16]. The results were in good agreement with theoretical estimations of the non-linear properties of various radio receiver front-ends with different designs. Finally, an automotive intermodulation radar for detecting and identifying vulnerable road users was proposed in [17].

However, none of the cited works reported a detailed design and characterization of the non-linear target. Moreover, an accurate modeling of the system, able to evaluate the performance of the radar as the used powers and frequencies and the distance from the target vary, is also lacking. Moreover, little attention has been devoted to the study of the system components responsible for the intermodulation or to the effect of the system power supply.

Therefore, in this work, the design and characterization of a non-linear target is presented. The response of the target was simulated with the microwave office (MWO) software available inside the AWR design environment Version 22 by Cadence and a model of the intermodulation radar in the presence of a non-linear target was implemented using the VSS software available inside AWR Version 22. Finally, the intermodulation produced by connectorized and antenna-equipped devices was studied using two specific experimental setups. This study will allow the identification of the system components producing intermodulation, and evaluation of the frequency response of various transceivers, both in normal operating conditions and when switched off.

## 2. Target Design

The non-linear target was realized using a HSMS2850 Schottky diode [18]. The parameters of the Spice model of the diode suggested by the manufacturer were B_V_ = 3.8 V, C_JO_ = 0.18 pF, E_G_ = 0.69 eV, I_BV_ = 3 × 10^−4^ A, I_S_ = 3 × 10^−6^ A, N = 1.06, R_S_ = 25 Ω, and V_J_ = 0.35 V, M = 0.5 [18]. Using the Keithley 236 source measurement unit, the current–voltage characteristic of the diode was measured. In order to fit the measured response with the Spice model, it was necessary to set R_S_ = 35 Ω and to add a resistance of 127 kΩ in parallel to the diode. Figure 1 shows the comparison between the measured I-V characteristic and the simulated ones with the initial parameters values (Default) and with the modified parameters (Modified).

Thanks to the model corrections, a very good agreement between measurements and simulations was obtained. The non-linear target was designed to operate at frequencies around 1 GHz with incident powers of about 0 dBm, close to those expected in practical cases (see Section 6 of this manuscript). To realize the target, a reactive network was added to the Schottky diode. This network consists of a parallel inductor, necessary to maintain the zero-bias condition for the diode, and a series matching capacitor. Then, two tones were applied to port 1 at the frequencies of 0.995 GHz and 1.005 GHz, each with a power equal to 0 dBm. Finally, the network was optimized with MWO software, available inside the AWR design environment Version 22, in order to maximize the generation of third-order intermodulation and to minimize the two-tone reflection coefficient. For the solution of the non-linear problem and for the circuit optimization, the Cadence APLAC^®^ HB non-linear harmonic balance solver and the Hybrid Pointer solver, which combines different global and local search methods, were used; both are available within the MWO software.

The obtained circuit is shown in Figure 2a, while its physical realization on a Roger RO4003, 30 mills substrate is shown in Figure 2b.

## 3. Target Measurements

The first tests on the target were carried out using the experimental setup whose scheme is shown in Figure 3.

The same output power P_G_ (dBm) was set on the two generators (Agilent E4432B (Santa Clara, CA, USA) and Anritsu MG3681A (Atsugi, Japan)) at the frequencies low (f_L_) and high (f_H_). The two signals were then combined by the power splitter/combiner (MINI CIRCUIT ZN2PD2-63-S+ and sent to the target after passing through a directional coupler (MINI CIRCUIT ZGDC10-362HP). Due to the cable and power splitter losses, the power incident on the target was approximately 5 dB lower than P_G_. Through the coupled port (C = 10 dB) of the directional coupler, the signals generated by the target (reflections and intermodulation) were sent to the spectrum analyzer (SPA HP8594E) and acquired by a PC via the GPIB interface. Before studying the intermodulation produced by the target, the third-order intermodulation of the spectrum analyzer was investigated. The value of −70 dBc for two −30 dBm tones at the input of the SPA mixer, reported on the SPA specifications, was confirmed by measurements. Furthermore, the intermodulation produced by the generators was also tested. In fact, a fraction of the power generated by one source couples with the other through the power combiner and generates intermodulation. At the maximum power that can be delivered by the generators (P_G_ = 15 dBm), a third-order intermodulation power of −62 dBm was measured with the SPA, which, therefore, was more than 75 dB below the tone power.

The power spectrum generated by the designed target for P_G_ = 0 dBm, f_L_ = 0.995 GHz, and f_H_ = 1.005 GHz is reported in Figure 4. In this case, the central frequency f_c_ = (f_L_ + f_H_)/2 is equal to 1 GHz and the tone difference Δf = f_H_ − f_L_ is equal to 10 MHz. The figure shows the presence of the left third-order intermodulation products (f_IM3L_ = 2f_L_ − f_H_ = 0.985 GHz) with power P_IM3L_ = −30 dBm and of the right third-order intermodulation product (f_IM3H_ = 2f_H_ − f_L_ = 1.015 GHz) with power P_IM3H_ = −33 dBm. As highlighted in the introduction of this work, this effect is due to the termination impedances of the target at the difference frequency (f_H_ − f_L_) [2,3]. Moreover, the reflected f_L_ and f_H_ tones with power P_RL_= −12 dBm and P_RH_ = −12 dBm are also evidenced.

It is important to note that the measured intermodulations are not due to the spectrum analyzer, as the internal attenuator of the SPA was set to an attenuation value of 30 dB and therefore the power at the mixer input of the SPA was lower than the aforementioned limit value of −30 dBm. The third-order intermodulation was quantified through the power of the left third-order intermodulation (P_IM3L_ (dBm)). Moreover, the reflections were quantified through the return loss defined as the ratio between the incident and the reflected powers of the f_L_ tone. The P_IM3L_ and return loss trends measured as a function of the generator powers for the case f_c_ = 1 GHz and Δf = 10 MHz are reported in Figure 5a,b, respectively. On the same figures, the results of the simulations performed with the MWO software and using the circuit in Figure 2 are also reported. Both figures show a good agreement between measurements and simulations.

Similar measurements and simulations were carried out at the central frequencies of 750 MHz and 1500 MHz, and the results are reported in Figure 6 and Figure 7, respectively.

Figure 5, Figure 6 and Figure 7 show, for low incident powers, an increase in the power of the intermodulation of about 3 dB/dB, as expected from the theory [1,19] and highlighted in the paper introduction. The highest values of the P_IM3L_ are obtained around 0 dBm. For higher incident powers, the P_IM3L_ saturates and there are some deviations between the measured and the simulated values, probably due to the incorrect behavior of the Spice diode model available inside MWO. Finally, in all the analyzed cases, the best matching was obtained for an incident power of each tone around 0 dBm, equal to that at which the target was optimized.

## 4. Device Measurements

As previously mentioned, one of the possible applications of an intermodulation radar consists in the identification of electronic systems. To this end, the experimental setup for connectorized devices shown in Figure 3 was used to investigate the intermodulation at the ports of various devices present in the electronic systems, both switched on and switched off.

First, the response of the ZEM-4300 Mini-Circuit mixer [20] was evaluated. For these measurements, two tones were used at the frequency of 0.995 GHz and 1.005 GHz with P_G_ = 0 dBm. The three ports IF (Figure 8a), RF (Figure 8b), and OL of the mixer were tested with the other ports left open. The obtained power spectra show third-order intermodulation powers P_IM3L_ = −27 dBm, −24 dBm, and −24 dBm at the mixer’s OL, RF, and IF ports. The high intermodulation levels produced by the mixer are related to the double-balanced nature of this mixer, which is realized with four Schottky diodes.

A similar test was conducted on the switched-off ZX60-H242 Mini-Circuit amplifier [21]. Figure 9a,b show the spectrum of the reflected power when the amplifier is excited at the input port with the output port left open and when the amplifier is excited at the output port with the input port left open, respectively. The power of reflected fundamental tones and the third-order intermodulation products were P_RL_ = −12 dBm and P_IM3L_ = −43 dBm when the input port is excited (see Figure 9a), and P_RL_ = −10 dBm and P_IM3L_ = −46 dBm when the output port is excited (see Figure 9b).

To further characterize this device, the power of each of the two incident tones was varied between −30 dBm and 10 dBm. Figure 10 shows P_RL_ and P_IM3L_ as a function of the incident power at the input (Figure 10a) and output (Figure 10b) ports, respectively. In each figure, the switched-off and switched-on (+5.5 V DC) behaviors are reported.

Figure 10a,b shows that the reflected power grows linearly (1 dB/dB) with the incident power while the power of the intermodulation products grows at about 3 dB/dB. Moreover, when the device is switched on, there is, at both ports, a reduction in the reflected power and in the intermodulation products. These results indicate that when the amplifier is switched off, the active components inside it are working in interdiction and are therefore in a highly non-linear state, while, when switched on, the working point moves in a more linear region, particularly with regards to its output.

Finally, a configuration always present in transceiver systems, consisting of the cascade of an amplifier and a mixer, was analyzed. When the amplifier is turned on and the two tones are injected at its input, the presence of strong intermodulation is highlighted (see Figure 11a). For comparison purposes, Figure 11b shows the response of the switched-on amplifier closed on a 50 Ω load. In particular, when the amplifier is turned on and closed on a 50 Ω load, the level of intermodulation is lower than −60 dBm, while, if the amplifier output is connected to a mixer, the strong non-linearities of this device are visible at the cascade input, producing intermodulation levels greater than −40 dBm.

## 5. Antenna Target Measurements

The block diagram of the radar used for measurements on antenna-equipped devices is reported in Figure 12. The two signal generators and the combiner are the same as those previously used (see Figure 3). Vivaldi antennas proposed in [22] were used as radiating elements. These antennas are well matched in the 700 MHz–3 GHz band, with a gain of about 4 dBi.

In order to evaluate the direct coupling between antennas 1 and 2, a measurement was carried out without antenna 3 and setting their distance DA = 30 cm. Antenna 1 was fed with two tones at the frequencies of 0.995 GHz and 1.005 GHz with a power of 0 dBm, and the spectrum analyzer (SPA) was linked to the antenna 2 port. In this way, the two tones’ measured power was about −27 dBm, and therefore the coupling between the two antennas resulted in about 27 dB. Then, the SPA was moved to the port of antenna 3 placed at a distance DT = 30 cm from the other two antennas (see Figure 12). Under these conditions, the SPA measured a power of −19 dBm, which represents the power that will impinge on the target. In practice, this means that the channel between the antenna port 1 and antenna port 3 in these conditions attenuates about 19 dB.

Finally, maintaining DA = 30 cm and DT = 30 cm, the target described in Section 2 was linked to the antenna 3 port and the SPA was linked to the antenna 2 port, as in Figure 12. In this case, the power spectrum measured by the SPA is reported in Figure 13a. With a power of tones impinging on the antenna 1 port equal to 0 dBm, the measured spectrum shows the presence of third-order intermodulation tones with a power P_IM3L_ = −73 dBm. By increasing the power of the two tones to 10 dBm, the power of the intermodulation rises to P_IM3L_ = −52.7 dBm (Figure 13b). In both figures, the power of the reflected tones is the incident one reduced by the 27 dB coupling between the two antennas.

## 6. Model of the Setup for Connectorized and Antenna-Equipped Devices

The experimental setup for connectorized devices shown in Figure 3 was modeled within the VSS software available in the Cadence AWR Version 22 CAD (see Figure 14).

VSS works in the time domain on sampled signals and is able to perform the analysis of a system constituted by the cascade of various blocks. Within VSS, it is possible to import the non-linear model of a device studied in the frequency domain with MWO. In particular, the Target block in Figure 14 is constituted by the circuit in Figure 2a with the addition of a circulator at the input (port 1) used to measure the target reflected power. As evidenced in Figure 14, the output of the circulator is sent to the test point (TP). Figure 15 shows the power spectrum of the signal reflected by the target and measured at the test point (TP) of Figure 14.

The same figure also reports the simulated power spectrum of the reflected signal obtained with MWO for the same frequencies and powers of the incident tones and using the circuit reported in Figure 2a. As can be seen, the two simulations provide practically the same results.

Then, the setup for antenna-equipped devices, reported in Figure 12, was modeled within VSS as shown in Figure 16. For the antenna component used in the model, a gain equal to 4 dBi was considered and, regarding the path distance (Dist) between transmitting and receiving points, the following expression was used $Dist=sqrt\left({\left(\frac{DA}{2}\right)}^{2}+{\left(DT+LA\right)}^{2}\right)$, where DA is the distance between antenna 1 and antenna 2, DT is the distance between the antennas mouth planes, and LA is the antenna length (see Figure 12). In the VSS model, the coupling between antenna 1 and antenna 2 was also modeled with the DCOUPLER and COMBINER blocks, where the DCOUPLER presents a coupling equal to the measured one (27 dB). Finally, to take into account the structure of the target implemented in VSS, antenna 3 was modeled using two blocks, one for transmission and the other for reception.

The experimental setup in Figure 12 was simulated by considering two tones with frequencies f1 = 0.995 GHz and f2 = 1.005 GHz with a power of 15 dBm. Moreover, DA = 30 cm, DT = 30 cm, and LA = 30 cm were used, thus reproducing the same experimental condition with which the results reported in Figure 13b were obtained. The simulated spectrum, evaluated at the point P_SPA_ in Figure 16, is reported in Figure 17 together with the measured one. A very good agreement between the two signals can be observed both with reference to the reflected fundamental tones and the intermodulation products.

Finally, Figure 18 shows the power (P_T_) of the two tones impinging on the target at the P_T_ point in Figure 16 and the power (P_IM3L_) of the third intermodulation tone at the P_SPA_ point as a function of the antenna target distance when the power of each of the two generated tones is P_G_ = 15 dBm. A very good agreement between the measurement performed with the experimental setup in Figure 12 and simulation with the model in Figure 16 can be observed.

It is worth noting that, with the SPA noise level equal to about −90 dBm, the maximum distance at which the realized non-linear target can be detected with a signal-to-noise ratio SNR = 10 dB is about 1.2 m. If the generated power (P_G_) is increased to 30 dBm (1 W), a SNR = 10 dB is guaranteed by the model simulations up to a distance of about 5 m between the radar antenna and the non-linear target.

## 7. Transceiver Measurements

This section reports the results of measurements performed with the experimental setup shown in Figure 19, in which the power of the two tones with Δf = 2 MHz was set equal to P_G_ = 15 dBm. In particular, measurements were performed on a WS7200 modem–router [23], on a Tom-Tom 4N00.004.2 navigator [24], and on a BF-88E two-way radio [25].

A first set of measurements was carried out on the Huawei WS7200 modem–router (Figure 20a) switched on and switched off, placed at a distance DD = 20 cm from the antennas with DA = 10 cm. A frequency sweep was performed in the band between 2000 MHz and 3000 MHz, obtaining P_IM3L_ as a function of the frequency reported in Figure 20b. The figure shows that when the router is switched on, there is an increase in the power of some intermodulation products but a strong reduction around the 2400 MHz frequency.

Then, measurements were performed on the “TomTom one” portable GNSS vehicle navigator (see Figure 21a), switched on, and by setting DA = 20 cm and DD = 15 cm. Figure 21b shows P_IM3L_ achieved by varying the central frequency of the two stimulating tones between 1450 and 1700 MHz around the central frequency of the GNSS L1 band (1575 MHz). Two repetitions (1 and 2) of the same measurement were performed.

Finally, the BF-88E two-way radio (see Figure 22a) placed at a distance DD = 15 cm from the antennas was investigated, both switched on and switched off. For these measurements, two antipodal Vivaldi antennas with a 300–700 MHz band [26] were used with DA = 10 cm. Figure 22b shows the frequency behavior of P_IM3L_ in the 400 MHz and 550 MHz region, both in the switched-on and switched-off mode. In this case, intermodulation is generated in a narrow bandwidth between 450 MHz and 500 MHz, and the level of the intermodulation products is almost the same in the two conditions.

Figure 23 shows a comparison among the powers of the intermodulation products of the three analyzed transceivers. The figure evidences the possibility of detecting the presence of a transceiver based on its peculiar frequency response. Finally, we want to highlight that in all the measurements performed on the transceivers, the difference between the main tone power level and the power level of the intermodulation produced by the transceiver was significantly lower than the 75 dB limit of the intermodulation produced by the generators.

## 8. Conclusions

The design and characterization of a non-linear target were performed using the AWR design environment Version 22 by Cadence software. This target was characterized using two experimental setups for intermodulation measurements on connectorized and antenna-equipped devices. Both setups were modeled using the VSS software available inside AWR Version 22. The comparison between measurements and simulations on the connectorized and antenna-equipped target showed a very good agreement. It has to be noted that a target similar to the one proposed in this work was presented by Mishra, Li in reference [6], where the authors designed a target based on a diode scattering parameter model, which was studied with AWR’s MWO software. The experimental setup used for the measurements was very similar to the one used in this work. The authors highlighted experimentally the presence of intermodulation products, but no comparison was made between the simulations and the measurements. The realized target could be used in many applications. For example, by placing it on an object and evaluating the phase of the received intermodulation signal, it is possible to study the distance of the object (carrying out two or more measurements as one of the two frequencies varies) and its position (carrying out two or more measurements as the position of the antennas varies). Therefore, the availability of a simple, well-characterized, non-linear target is crucial in many applications and for designing an intermodulation radar system and predicting its response.

Intermodulation measurements were performed on connectorized devices present inside electronic systems and on various transceivers available on the market. This study evidenced that the non-linearities of devices such as amplifiers and mixers present in almost all electronic systems are visible at their access ports, even when the device is turned off. Moreover, the ability of an intermodulation radar to remotely detect the presence of a particular transceiver, both switched on and off, thanks to the specific frequency response of its intermodulation products, has been highlighted.

## Figures and Tables

**Figure 1 sensors-24-01433-f001:**
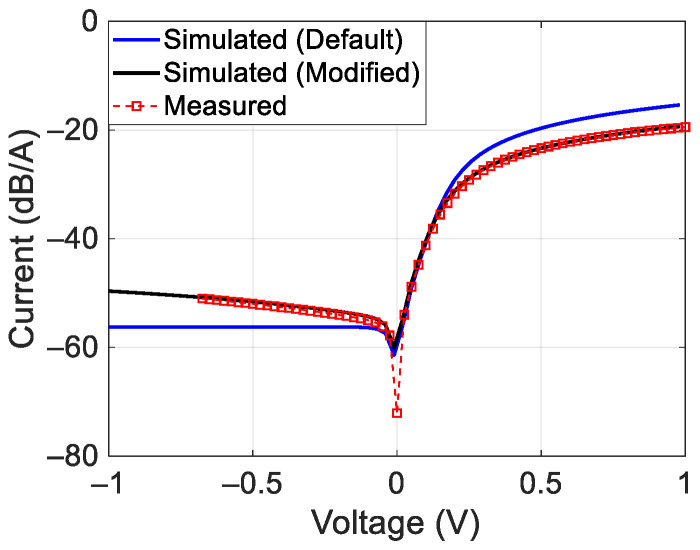
Measured and simulated I–V characteristic of the HSMS2850 Schottky diode.

**Figure 2 sensors-24-01433-f002:**
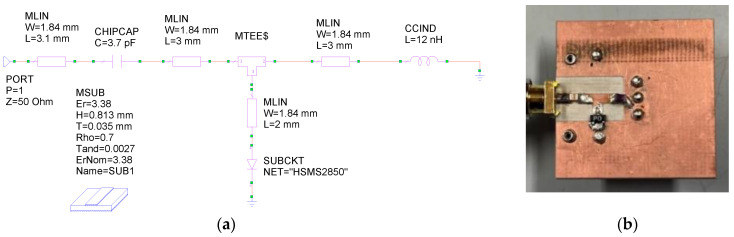
Scheme of the realized target (**a**) and its physical realization (**b**).

**Figure 3 sensors-24-01433-f003:**
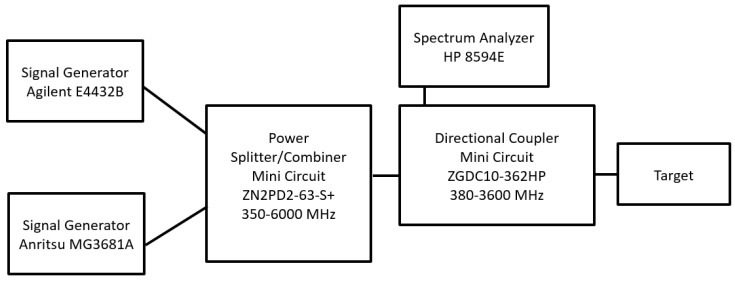
Scheme of the experimental setup for connectorized devices.

**Figure 4 sensors-24-01433-f004:**
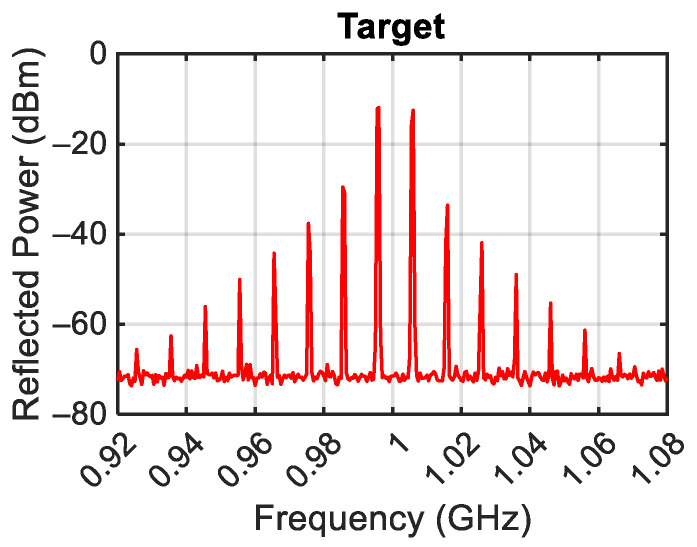
Measured target response.

**Figure 5 sensors-24-01433-f005:**
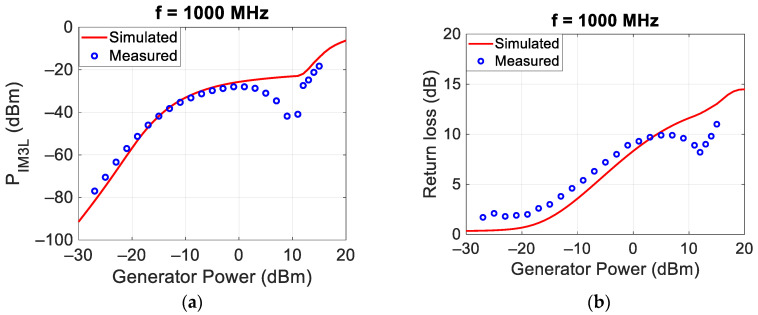
Comparison between measured and simulated P_IM3L_ (**a**) and return loss (**b**) at 1000 MHz as a function of the power of each of the two generators.

**Figure 6 sensors-24-01433-f006:**
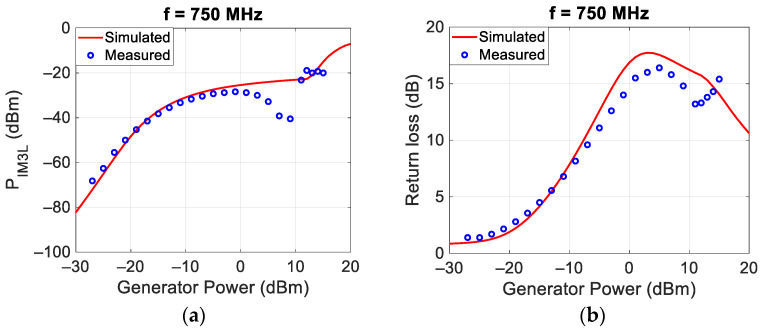
Comparison between measured and simulated P_IM3L_ (**a**) and return loss (**b**) at 750 MHz.

**Figure 7 sensors-24-01433-f007:**
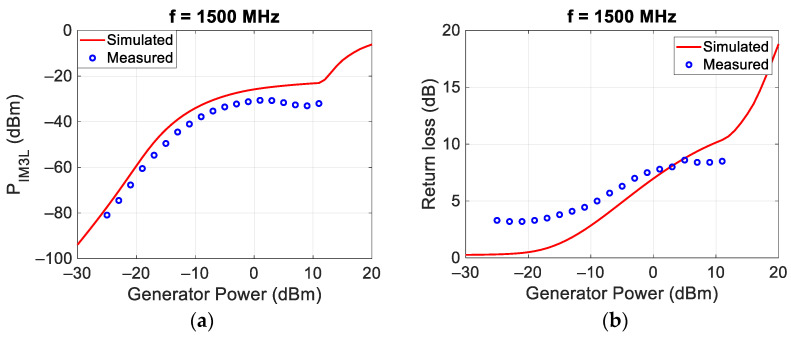
Comparison between measured and simulated P_IM3L_ (**a**) and return loss (**b**) at 1500 MHz.

**Figure 8 sensors-24-01433-f008:**
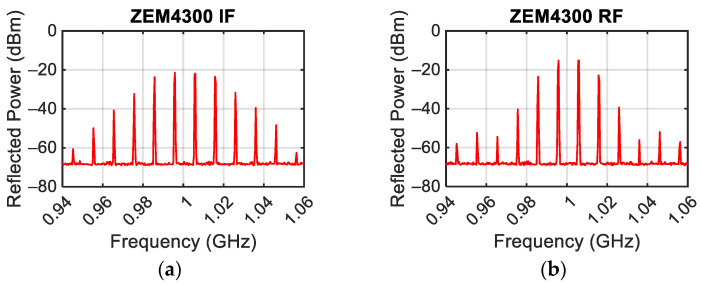
Power spectrum measured by exciting the IF port of the mixer (**a**), and power spectrum measured by exciting the RF port of the mixer (**b**) with a power of each tone equal to 0 dBm.

**Figure 9 sensors-24-01433-f009:**
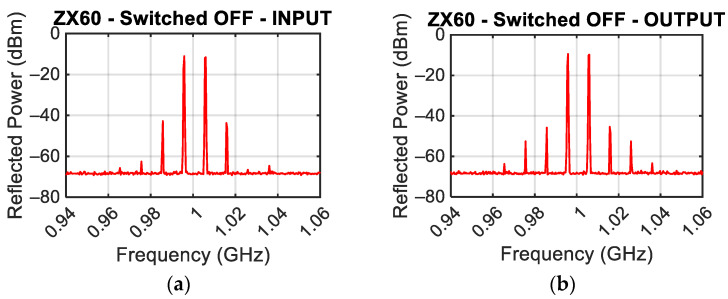
Power spectrum measured by exciting the amplifier input, (**a**) and power spectrum measured by exciting the amplifier output (**b**) with a power of each tone equal to 0 dBm.

**Figure 10 sensors-24-01433-f010:**
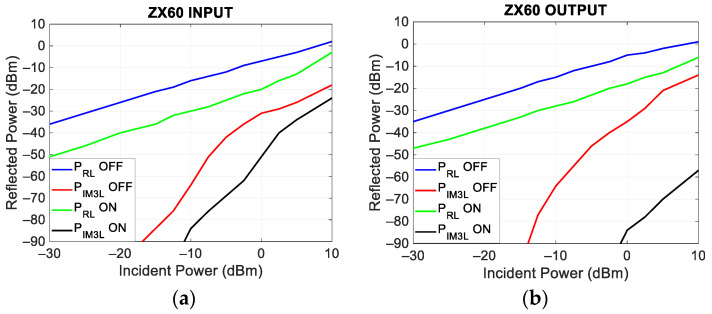
Reflected power as a function of incident power, at the ZX60 amplifier input (**a**) and output (**b**) ports. On the same plots, the measured responses with the amplifier switched on and switched off are reported.

**Figure 11 sensors-24-01433-f011:**
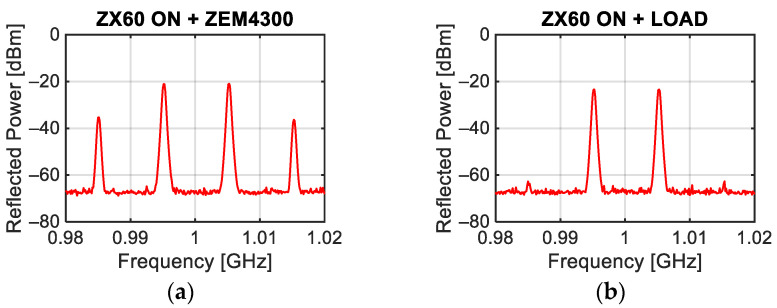
Power spectrum measured by exciting the input of the ZX60 amplifier whose output is connected to the RF port of the ZEM4300 mixer (**a**) or closed on a 50 Ω load (**b**) with a power of each tone equal to 0 dBm.

**Figure 12 sensors-24-01433-f012:**
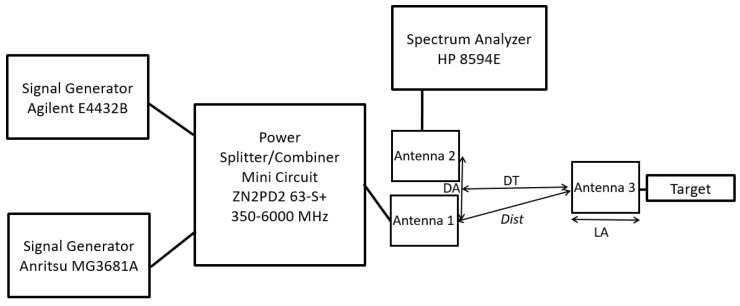
Scheme of the setup for measurement on antenna-equipped devices.

**Figure 13 sensors-24-01433-f013:**
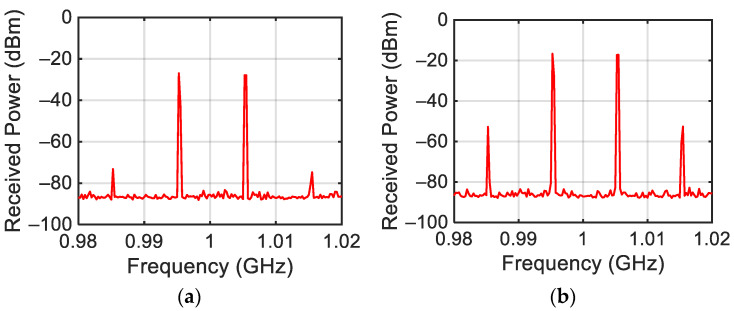
Measured intermodulation with two tones impinging power equal to 0 dBm (**a**) and 10 dBm (**b**).

**Figure 14 sensors-24-01433-f014:**
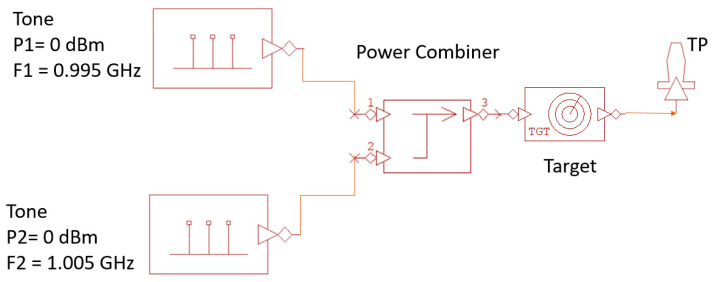
VSS model of the setup in Figure 3.

**Figure 15 sensors-24-01433-f015:**
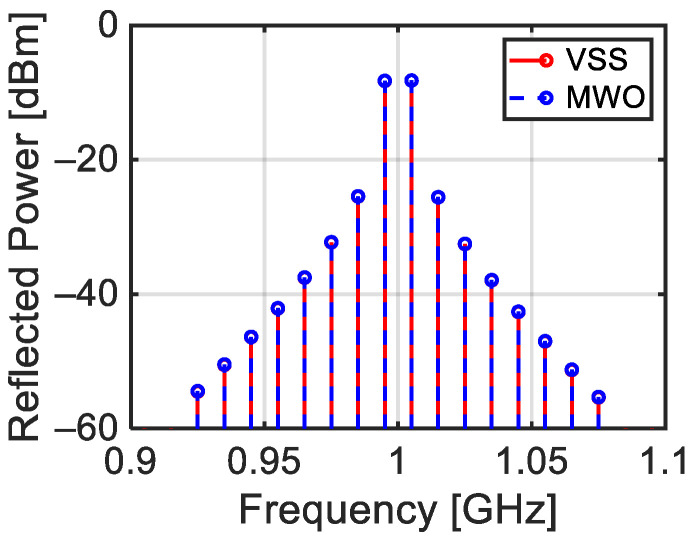
Comparison between the spectra achieved with VSS for the circuit in Figure 14 and with MWO for the circuit in Figure 2a.

**Figure 16 sensors-24-01433-f016:**
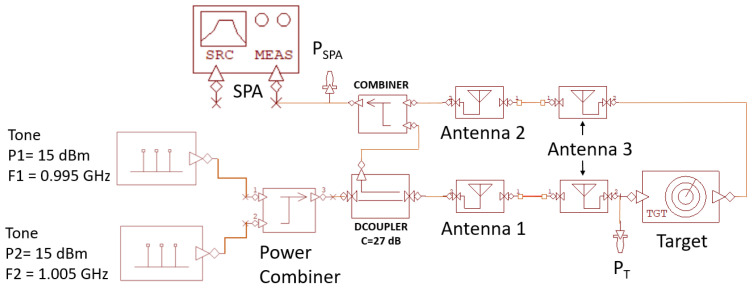
VSS model of the experimental setup in Figure 12.

**Figure 17 sensors-24-01433-f017:**
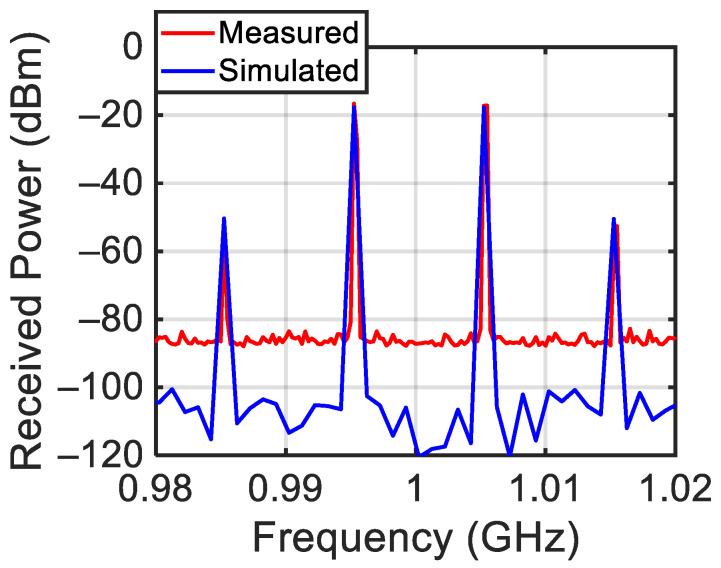
Comparison between the spectrum measured with the setup in Figure 12 and that simulated with the VSS model in Figure 16.

**Figure 18 sensors-24-01433-f018:**
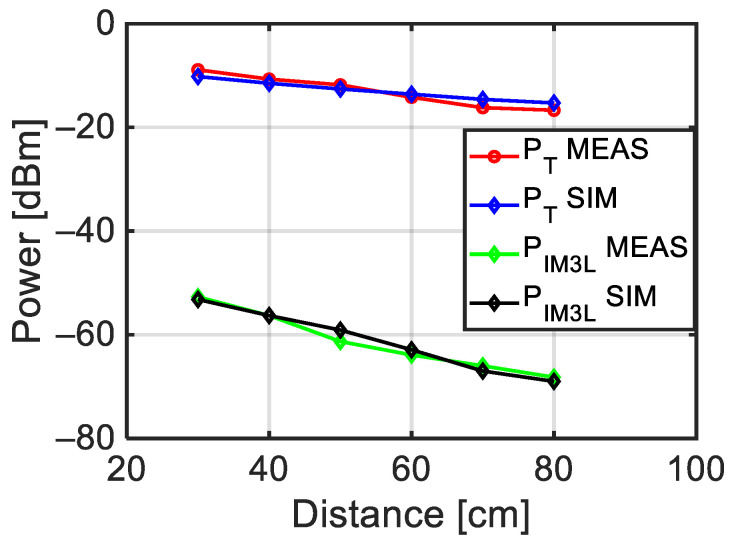
Power of the tones incident on the target and of the intermodulations produced by the target, as a function of the radar-target distance, measured with the setup in Figure 12. Comparison with simulations performed with the VSS model in Figure 16.

**Figure 19 sensors-24-01433-f019:**
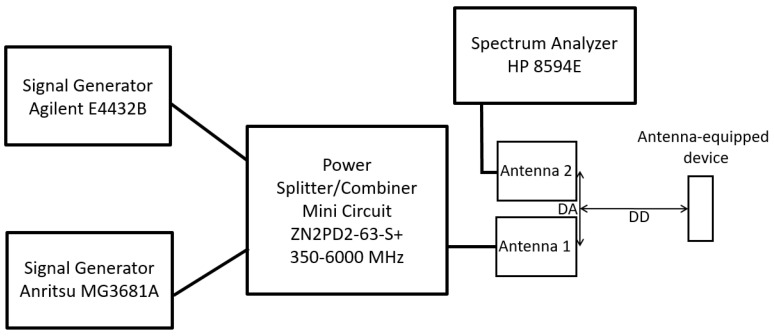
Scheme of the setup for measurement on transceivers.

**Figure 20 sensors-24-01433-f020:**
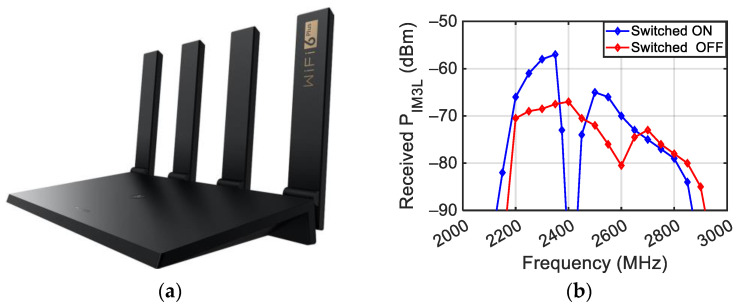
Picture of the WS7200 modem–router (**a**) and measured third-order intermodulation with two tones’ impinging power equal to 10 dBm (**b**).

**Figure 21 sensors-24-01433-f021:**
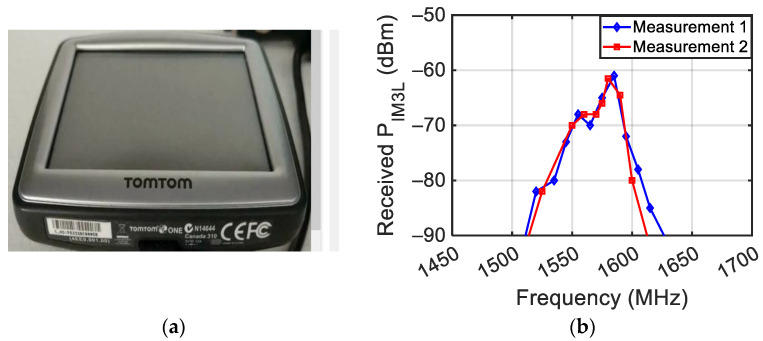
Picture of the TomTom one portable GNSS vehicle navigator (**a**) and measured third-order intermodulation with two tones’ impinging power equal to 10 dBm (**b**).

**Figure 22 sensors-24-01433-f022:**
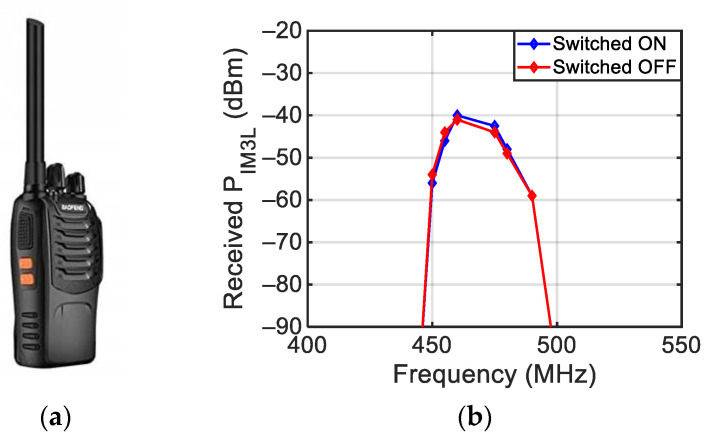
Picture of the BF-88E two-way radio (**a**) and measured third-order intermodulation with two tones’ impinging power equal to 10 dBm (**b**).

**Figure 23 sensors-24-01433-f023:**
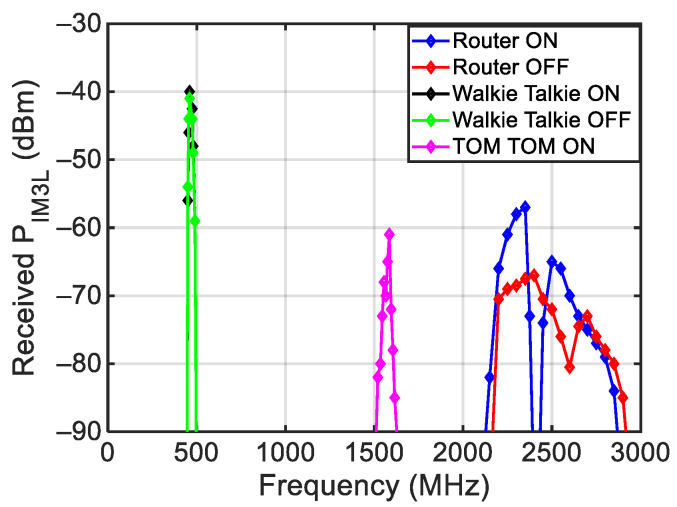
Comparison among the measured third-order intermodulation of the three considered transceivers.

## Data Availability

Data are contained within the article.

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
