# Peer review of "An Intermodulation Radar for Non-Linear Target and Transceiver Detection"

_sensors, 2024, doi:10.3390/s24051433_

Round 1

Reviewer 1 Report

Comments and Suggestions for Authors

The authors describe a work related to nonlinear radars. As a matter of fact, t  they basically made simulations and measurement to detect siganls leaked associated to third-order nonlinear  characteristic of electronic devices. The motivation for the work is interesting, but I recommend to give more information about it. The simulations were performed with AWR software and measurements were performed with a Spectrum analyzer.

General comments

* In terms of simulations of measurement, the authors used  basic/classic methods

*It would be interesting to the readers if details about simulations were give (which kind of nonlinear simulations was used, etc.)

* Concerning the references, one can easily observe that they are very limited, since the authors cite several papers of two main authors/teams. Even though we can find a longer list refering to books, manuals, datasheet, etc. I recommend to improve the reference list as a consequence.

*The authors did a prototype with diodes (obviously a nonlinear device) but did not relate the results with any results or general models.

*Concerning the results, it is very important that the authors perform a comparison with other relevant works.

*Still with respect to the results,  they are  not conclusive, since they cannot be generalized, due to lack of exhaustive theorectical study and very limited number os measurements. I recommend to offer theorectical analisys that allow to understand the differences between on/off behaviors, power limits, etc.

* Since a mixer is not the first block on a receiving chain, I suggest the authors to make measurements considering it after a LNA, at least. Possibly, the LNA blocks the back propagation of IM3 signals to the antenna.

* The modelling part is really  limited and it was not even used in the final part.

* Finally, I would recommend to lin the results with recommendations for radar devellopers/users. What is the main outcome of the research?

Author Response

  1. In terms of simulations of measurement, the authors used basic/classic methods

In this work, by using simulations and experimental comparisons, it has been developed a non-linear target that has been then used to characterize two experimental setups for measurements on components connected or equipped with antennas. Moreover, the possibility of detecting transceiver systems both switched on or off by stimulating the generation of intermodulation tones has been highlighted. For these studies it was not necessary to use innovative simulation and measurement methods. Possible developments of the work include the use of more accurate non-linear models to achieve a better agreement between simulations and measurements on the target and the study of the phase of the intermodulation products to obtain the position of the target. These developments will require more innovative measurement models and techniques.

  1. It would be interesting to the readers if details about simulations were give (which kind of nonlinear simulations was used, etc.)

Thank for this comment. For the solution of the nonlinear problem and for the circuit optimization, the Cadence APLAC® HB nonlinear harmonic balance solver and the Hybrid Pointer solver, which combines different global and local search methods, were used, both available within the MWO software. This consideration has been added into the text of the revised version of the manuscript.

  1. Concerning the references, one can easily observe that they are very limited, since the authors cite several papers of two main authors/teams. Even though we can find a longer list referring to books, manuals, datasheet, etc. I recommend to improve the reference list as a consequence.

Although the literature on nonlinear radars is very extensive, as reported in the article and highlighted by the reviewer, only 3-4 groups have reported results obtained with intermodulation radars. Therefore, in the revised version of this work, in order to better present the literature of intermodulation radars, the activities of these groups were better described by adding further references. Furthermore, in the introduction, space was given to the general theory of intermodulation and to its characteristics.

  1. The authors did a prototype with diodes (obviously a nonlinear device) but did not relate the results with any results or general models.

The target realised in this work was designed and studied using the MWO software where the diode is modelled using the so-called Spice model in which the diode current is linked exponentially to the voltage across the diode for low voltages and then it is linearized for high voltages. With this model the software is able to reconstruct the intermodulation products quite accurately as highlighted by the comparisons between theory and experiments reported in figures 5,6,7 of this work. Classic analytical models such as the Volterra or Taylor series are able to predict the behaviour of a non-linear circuit but only for low or medium power signals as they do not take into account the saturation processes which are instead present in the MWO Spice model. The development of a general nonlinear model that also takes saturation into account is particularly challenging but outside the objectives of the present work.

  1. Concerning the results, it is very important that the authors perform a comparison with other relevant works.

A target similar to the one proposed in this work was presented by Mishra, Li in reference [6] of this work. In [6] the Authors designed the target based on a diode scattering parameter model which was studied with AWR's MWO software. The experimental setup used for the measurements was quite similar to the one used in this work for measurements on connectorized devices. The Authors highlighted experimentally the presence of intermodulation products but no comparison was made between the simulations and the measurements. As highlighted in the text of our article, the good agreement between measurements and simulations performed on the target represents a strong point of our work which has allowed us to use the target also for the characterization of the system for the measurement on transceivers. The comparison with [6] has been added in the conclusions of the revised version of the paper.

  1. Still with respect to the results, they are not conclusive, since they cannot be generalized, due to lack of exhaustive theoretical study and very limited number of measurements. I recommend to offer theoretical analysis that allow to understand the differences between on/off behaviours, power limits, etc.

With reference to the measurements carried out on amplifiers and mixers, our objective was to highlight the possibility of detecting an electronic system using an intermodulation radar. In particular, we wanted to see if the nonlinearities present inside the device were also visible at the external ports. To our knowledge, this type of evaluation has never been reported in the literature. Furthermore, the performed measurements highlighted that the nonlinearities of the device are visible even when the device is turned off. Clearly, the powers of third-order intermodulation products obtained by measuring amplifiers and mixers other than those used in this work will be different. However, in this work we have defined a general procedure for measuring device intermodulation. Furthermore, if the circuit model of a device was accessible, the same results could be obtained through simulations.

As regards the difference in behaviour of the amplifiers between the on and off states, as reported in the text of the article, this is due to the fact that, when an amplifier is off, the active devices inside it are working in interdiction and therefore in a highly nonlinear region. vice versa, when the amplifier is turned on the active devices work in the region of maximum linearity and therefore produce low levels of intermodulation. A general theoretical model would be very interesting to develop but is a challenging task and, in any case, outside the objectives of this work.

  1. Since a mixer is not the first block on a receiving chain, I suggest the authors to make measurements considering it after a LNA, at least. Possibly, the LNA blocks the back propagation of IM3 signals to the antenna.

We thank the reviewer for this comment which allows us to evidence an important point. In performing the measurements requested by the Reviewer we highlighted that when the amplifier is turned on, if it is closed on the RF port of the ZEM-430 mixer, strong intermodulation is generated at its input port, while if the amplifier is turned on and closed on a 50 ohm load the intermodulation is practically negligible. These measures were added to the text of the revised article and commented.

  1. The modelling part is really limited and it was not even used in the final part.

In the final part of the article, measurements on transceiver systems are reported. For these commercial systems it is not possible to have detailed information on their electronic circuitry, therefore it is not possible to use the proposed VSS model. Furthermore, in this part of the work we were more interested in verifying whether the proposed intermodulation radar is able to identify the presence of turned-on or turned-off transceiver systems.

  1. Finally, I would recommend to link the results with recommendations for radar developers/users. What is the main outcome of the research?

One of the main indications for those who want to develop a harmonic radar is to have a simple, well-characterised non-linear target available. The use of an intermodulation radar model based on system simulators or equivalently on analytical models is also fundamental for designing the radar and predicting its responses. Finally, it was highlighted that the frequency response of the intermodulation products of a transceiver system can be used to detect the presence of a transceiver. These points have been highlighted throughout the work and also reported in the conclusions.

Reviewer 2 Report

Comments and Suggestions for Authors

Please include a section describing how the nonlinear tones were removed from the transmitter before radiating out towards the target.

Please include X-parameter measurements of the nonlinear target as a function of distance from the transmitter.

Please include a discussion if the matching circuits  were matched to 50 ohms and how does the matching, affects the nonlinear tone generation.

Author Response

  1. Please include a section describing how the nonlinear tones were removed from the transmitter before radiating out towards the target.

We agree with the reviewer that the problem of intermodulation produced by the generators but also by the spectrum analyser (SPA) is one of the most critical problem of an intermodulation radar. With reference to the spectrum analyser, the manufacturer provides an intermodulation level of -70 dBc when two tones with a power of -30 dBm are applied to the input of the mixer present inside the SPA. This limit was experimentally verified and has been satisfied in all the measurements carried out also thanks to the use of the controllable attenuator present inside the SPA before the mixer. Concerning the generators, in our experimental system their maximum output power is 15 dBm. At this power level the combiner output was sent to the spectrum analyser and the presence of intermodulation with power of -62 dBm was highlighted. So the intermodulation level produced by the generators is more than 75 dB below the power of the two main tones. Also this condition has been satisfied in all measurements carried out in this work. This last consideration has been added in the revised version of the paper.

  1. Please include X-parameter measurements of the nonlinear target as a function of distance from the transmitter.

X-parameters are used for characterizing the amplitudes of harmonics generated by nonlinear components under large input power levels. Measurements of the X parameters can be performed using the PNA-X family of network analyzers from the Keysight company. The main advantage of using these network analyzers is that phase and amplitude measurements can be carried out in a very short time as a function of the frequency. However, for the application of our interest the information on the amplitude is sufficient and rapid measurements as a function of the frequency can be achieved by controlling the instruments through their GPIB port. In the developments of this work we want to use intermodulation radars for target distance and position measurements. In this case, information on the intermodulation phase will be fundamental and a first set of measurements will be conducted by leasing a PNA-X analyzer from the Keysight company. With reference to distance-dependent measurements, in section 6 of this work, they were conducted with the proposed intermodulation radar equipped with a Vivaldi antenna and it was seen that with the available powers it is possible to detect the realized target up to distances of approximately 80 cm. The good agreement between simulations and measurements obtained in this case makes us confident in the accuracy of our experimental setup.

  1. Please include a discussion if the circuits were matched to 50 ohms and how does the matching, affects the nonlinear tone generation.

In the revised version of this work, measurements have been added which show the spectra of the power reflected by an amplifier turned on as the load varies. Completely different behaviors have been highlighted when the amplifier is closed on a mixer or on a 50-ohm load. These results have been shown and commented in the revised version of the work.

Round 2

Reviewer 1 Report

Comments and Suggestions for Authors

The authors provided a revised version of the manuscript together with satisfactory answers to all  questions.

The paper was adequately improved, in my opinion and can be accepted in the present form.